# Novel *CRYGC* Mutation in Conserved Ultraviolet-Protective Tryptophan (p.Trp131Arg) Is Linked to Autosomal Dominant Congenital Cataract

**DOI:** 10.3390/ijms242316594

**Published:** 2023-11-22

**Authors:** Flora Delas, Samuel Koller, Silke Feil, Ivanka Dacheva, Christina Gerth-Kahlert, Wolfgang Berger

**Affiliations:** 1Institute of Medical Molecular Genetics, University of Zurich, 8952 Schlieren, Switzerland; flora.delas@uzh.ch (F.D.); koller@medmolgen.uzh.ch (S.K.); feil@medmolgen.uzh.ch (S.F.); 2Department of Ophthalmology, Cantonal Hospital of St. Gallen, 9007 St. Gallen, Switzerland; ivanka.dacheva@kssg.ch; 3Department of Ophthalmology, University Hospital of Zurich, 8091 Zurich, Switzerland; 4Neuroscience Center Zürich (ZNZ), University of Zurich and ETH Zurich, 8006 Zurich, Switzerland; 5Zurich Center for Integrative Human Physiology (ZIHP), University of Zurich, 8006 Zurich, Switzerland

**Keywords:** congenital cataract, *CRYGC*, crystallin, whole exome sequencing, conserved tryptophan, UV damage

## Abstract

Congenital cataract (CC), the most prevalent cause of childhood blindness and amblyopia, necessitates prompt and precise genetic diagnosis. The objective of this study is to identify the underlying genetic cause in a Swiss patient with isolated CC. Whole exome sequencing (WES) and copy number variation (CNV) analysis were conducted for variant identification in a patient born with a total binocular CC without a family history of CC. Sanger Sequencing was used to confirm the variant and segregation analysis was used to screen the non-affected parents. The first de novo missense mutation at c.391T>C was identified in exon 3 of *CRYGC* on chromosome 2 causing the substitution of a highly conserved Tryptophan to an Arginine located at p.Trp131Arg. Previous studies exhibit significant changes in the tertiary structure of the crystallin family in the following variant locus, making *CRYGC* prone to aggregation aggravated by photodamage resulting in cataract. The variant can be classified as pathogenic according to the American College of Medical Genetics and Genomics (ACMG) criteria (PP3 + PM1 + PM2 + PS2; scoring 10 points). The identification of this novel variant expands the existing knowledge on the range of variants found in the *CRYGC* gene and contributes to a better comprehension of cataract heterogeneity.

## 1. Introduction

Congenital cataract (CC), referring to any light scattering due to clouding of the crystalline lens detected at birth, is one of the leading causes of treatable childhood blindness and amblyopia worldwide [1,2,3]. It affects one to nine newborns per 10,000 live births globally [4]. Approximately 50% of CCs are inherited [5]. Inherited cataracts can phenotypically be distinguished by localization (i.e., polar, nuclear, lamellar, cortical, total), type of opacity (i.e., solid, pulverulent, blue dot, crystalline), and presence of sutural opacity (affecting y-sutures of the fetal lens nucleus), and are described accordingly: anterior polar, posterior polar, lamellar, cortical, nuclear, aculeiform, total, pulverulent, cerulean, or polymorphic cataracts [6]. Inherited CC may manifest independently (70%), with other ocular abnormalities (e.g., microphthalmia being the most common) (15%), or in conjunction with other systemic findings i.e., syndromic (15%) [7,8]. They are predominantly inherited in an autosomal dominant manner, therefore particularly penetrant, and display an extensive genetic and phenotypic heterogeneity; thus, it challenging to establish a genotype–phenotype correlation in CC [8,9]. The detection efficiency of genetic variants in familial and sporadic cataracts varies greatly. Panel-based sequencing shows a detection rate of around 75% in familial cases and ranges from 26% to 68% in sporadic cases [10,11]. Unknown genetic and nongenetic factors contribute to sporadic cases [11]. Whole exome sequencing (WES) has been shown to offer a higher diagnostic yield compared to a panel-based analysis, and according to recent studies, WES represents the genetic test of choice rather than whole genome sequencing (WGS) [9,12]. To date, the Online Mendelian Inheritance in Man (OMIM) documented the identification of 49 loci and 37 genes associated with isolated CC (https://www.ncbi.nlm.nih.gov/omim/ (accessed on 9 July 2023)). These associated genes can broadly be grouped into cytoplasmic proteins (i.e., crystallins), membrane proteins (i.e., connexins, aquaporins), cytoskeletal proteins, and DNA/RNA-binding proteins (i.e., transcription factors) [13,14].

Crystallin proteins make up over 90% of the soluble human lens protein; they are non-renewable, thus unusually stable serving a lifetime, and play a pivotal role in maintaining lens transparency and the refractive index of the lens [15,16]. Numerous mutations in the 12 crystallin (*CRY*) genes have been identified, accounting for almost 50% of all autosomal dominant inherited cataracts in humans described thus far [14]. There are three groups of crystallin proteins, α-, β-, and γ-crystallins. α-crystallins are small heat shock proteins. They exert their chaperone function by binding to unfolded or damaged β- and γ-crystallins to prevent their aggregation, preserving lens transparency [13]. β- and γ-crystallins function as structural proteins and contain Greek key domains as secondary protein structures [13]. A primary distinction between β- and γ-crystallins lies in their ability to assemble into oligomers. While γ-crystallins solely occur to be monomeric, β-crystallins have the capacity to form various oligomeric structures, like homomers or heteromers, ranging from dimers to octamers [17]. It is known that the Greek key domains in γ-crystallin contain four highly conserved Tryptophan (Trp) residues (i.e., Trp43, Trp69, Trp131, and Trp157), crucial for both protein stability and enabling ultraviolet radiation (UV) absorption with minimal protein damage (as in protein aggregation), which maintains lens transparency, ensuring UV protection for the retina [17,18]. Extensive photodamage to Trp residues within β- and γ-crystallin has widely been implicated as a contributing factor in the development of age-related cataracts [19]. The four conserved Trp residues display an efficient fluorescence quenching mechanism, which is understood to be an evolved property of protein folding, allowing UV absorption with minimal protein photodamage and delayed cataract formation [18].

In this study, we identified a de novo missense mutation in the crystallin γC (*CRYGC*) gene using WES, causing a substitution of one of the highly conserved Tryptophan at p.Trp131Arg in a patient with congenital nuclear cataract.

## 2. Materials and Methods

### 2.1. Patient

The index patient was identified through the cataract genetic study. The cataract genetic study at the Department of Ophthalmology, University Hospital Zurich, together with the Institute of Medical Molecular Genetics, University of Zurich, aims to characterize congenital cataracts by phenotype and genotype identification. Patients are identified and recruited through close collaboration with other ophthalmic centers in Switzerland. A detailed retrospective chart review was performed. In addition, the father received an undilated eye examination. Blood samples were collected from the index patient and both parents. The study adhered to the Good Clinical Practices and followed the guidelines of the Declaration of Helsinki [20]. Approval for genetic testing in human patients was awarded to the Institute of Medical Molecular Genetics by the Cantonal Ethics Committee of Zurich (Ref-No. 2019-00108). Written consent of the legal guardian of the patient was obtained. 

### 2.2. Genes of Interest

The gene list of Rechsteiner et al. 2021 [9] was expanded through the Human Gene Mutation Database (HGMD) as well as a current literature search (Appendix A). The gene list compiles cataract-associated candidate genes (syndromic and non-syndromic phenotypes), as well as cataract-associated genes in animal models.

### 2.3. Exome Sequencing and Analysis

We performed exome sequencing and analysis as previously described [9,21,22]. In brief, DNA was isolated from venous blood samples using the Chemagic DNA Blood Kit (Perkin Elmer, Waltham, MA, USA), fragmentation was executed using M220 Sonicator (Covaris, Woburn, MA, USA), and library preparation was performed using the IDT-Illumina TruSeq DNA Exome protocol (Illumina, San Diego, CA, USA and Integrated DNA Technologies, Coralville, IA, USA). Paired-end sequencing (2 × 75 bp) was executed using the NextSeq 550 instrument (Illumina, San Diego, CA, USA). The reads were aligned to the human genome (GRCh37) and variant calling was accomplished using Burrows–Wheeler Aligner (BWA) v0.7.17 on BaseSpace Onsite (Illumina). AlamutBatch version 1.10 (Interactive Biosoftware, Rouen, France) was used for variant annotation. Copy number variations (CNVs) within the genes of interest (Appendix A) were collected from exome coverage depth data (Sequence Pilot version 5.0; JSI Medical Systems GmbH, Ettenheim, Germany). Variants with a heterozygous allele frequency > 1%, a homozygous allele frequency > 0.01% (gnomAD heterozygous, and homozygous frequency of all populations; https://gnomad.broadinsitute.org/ (accessed on 12 June 2023)), and a Combined Annotation-Dependent Depletion (CADD) score ≤ 20 were discarded. 

### 2.4. Segregation Analysis

Segregation analysis was performed using Sanger sequencing as described in detail by Haug et al. (2021) [21]. In brief, the region of interest was amplified by PCR. Cycle sequencing was performed on the PCR products using BigDye™ Terminator V1.1 (Thermo Fisher Scientific, Waltham, MA, USA), followed by ethanol precipitation purification, and sequencing on a SeqStudio (Thermo Fisher Scientific, Waltham, MA, USA) capillary sequencer.

## 3. Results

### 3.1. Case Presentation

The patient was diagnosed with an abnormal red reflex at the age of 2.5 months and referred for further evaluation to the Cantonal Hospital of St. Gallen. Examination revealed a bilateral symmetrical dense and almost complete nuclear cataract not allowing fundus visibility. No associated ocular anomalies were diagnosed, particularly no microcornea or microphthalmia. The child was born on term, was developing well, and did not show any dysmorphic and/or systemic features. The family history did not reveal CCs, developmental and/or ocular anomalies, and consanguinity is not known. Lensectomy with primary posterior capsulotomy and anterior vitrectomy were performed in both eyes immediately after the diagnosis, within one week apart. The postoperative course in the left eye was complicated by increased inflammation despite intensive topical antibiotic and steroid treatment. After a second surgical intervention with detailed synechiolysis, no further complications occurred. Refractive correction was achieved by contact lens correction and bifocal glasses for near. Convergent strabismus and amblyopia in the right eye were diagnosed at the age of 13 months. Additionally, a secondary high-frequency pendular nystagmus to the left was described. Amblyopia treatment with patching therapy was initiated. Bilateral aphakic glaucoma was diagnosed at the age of 17 months and treated with topical anti-glaucomatous medication. The patient received cyclophotocoagulation in the right eye at three years of age. Intraocular pressure was controlled by topical medication until the last follow-up at the age of 10 years. The optic nerve displayed an increased cup-to-disc ratio (CDR) of 0.8 in the right eye. At this age, visual acuity (Snellen decimal) with contact lenses (right eye 10.75 diopters (dpt), left eye 14.5 dpt) and near correction of +6.0 dpt measured 0.2 and 0.3 at distance, and 0.3 and 0.4 at near, for the right and left eye, respectively. 

### 3.2. Segregation Analysis

An index patient WES data analysis of the selected genes (*n* = 278) revealed a total of 475 variants (12 deletions, 1 insertion, 6 duplications, and 456 substitutions). Due to the index patient being the only affected family member, we focused our filtering on de novo and recessive variants. Filtering for variants with heterozygous allele frequency ≤ 1% and homozygous allele frequency ≤ 0.01% revealed seven variants with a CADD score ≥ 20, one of which was revealed to be classified as pathogenic (PP3 + PM1 + PM2 + PS2; scoring 10 points) by means of the standard guidelines of interpretations according to the American College of Medical Genetics and Genomics (ACMG) [23] and highly damaging (score of 0.983 HumVar, sensitivity: 0.56; specificity: 0.94) according to PolyPhen2 (http://genetics.bwh.harvard.edu/pph/ (accessed on 9 June 2023)). The identified variant, located in exon 3 (c.391T>C) of *CRYGC* (RefSeq NM_ 020989.4), causes a protein change from a highly conserved Tryptophan across species (Figure 1) with a phyloP score of 7.02 (indicating a high level of evolutionary conservation) to Arginine in codon 131 (p.Trp131Arg) (Table 1). No mosaicism was found in either blood sample; however, germline mosaicism remains unknown. Due to absence of the variant in both parents, it is considered de novo and has been verified using Sanger Sequencing as indicated (Figure 2).

## 4. Discussion

The *CRYGC* protein, like all γ-crystallins, exhibits a distinctive structural arrangement with a two-domain β-structure, consisting of four Greek key motifs that are remarkably similar in their folding pattern, displaying a high degree of symmetry and strong stability consequently [24]. As indicated in Table 2, 41 disease-causing mutations have been identified in *CRYGC* thus far, all of which cause various types of CC with or without microphthalmia. Most *CRYGC* mutations display a severe disruption of protein stability and symmetry due to either a frameshift or stop gain mutation (Table 2). Chen et al. (2009) [18] revealed significant findings on the ability to effectively quench excited states through electrostatic interactions of four highly conserved Tryptophans (Trp 43, Trp69, Trp131, and Trp 157) on a protein basis, to be an evolved property of all γ-crystallins to maintain the tertiary structure as a form of UV protection. Thus far, only 14 *CRYGC* missense mutations have been published, none of which affect these highly conserved Tryptophans (Table 2).

Out of all crystallin families, only five mutations in conserved Tryptophans have been published thus far (Table 3). Wang et al. (2011) [59] reported the first human γ-crystallin mutation in one of the four conserved Trp residues, p.Trp43Arg in *CRYGD*, in a Chinese family with autosomal dominant nuclear CC, revealing notable alteration in the tertiary structure despite a lack of secondary structural changes, as well as protein aggregation upon UV radiation of the *CRYGD* mutant. Ji et al. (2013) [60], on the contrary, described a very similar x-ray structure between the wild-type *CRYGD* and the p.Trp43Arg mutant. Instead, a significant change in the stability and solubility behavior has been demonstrated, particularly in terms of protein folding and unfolding dynamics, being responsible for cataract formation (i.e., protein precipitation and aggregation) [60]. Interestingly, there is a link between the p.Trp43Arg *CRYGD* mutant and UV-damaged wild-type *CRYGD* (i.e., in age-related cataract), displaying similar precipitation dynamics in vitro [60]. Rao et al. (2013) [61] demonstrated that UV light, in the later stages of gestation of mouse fetuses, plays a significant role in activating melanopsin-expressing retinal ganglion cells, thus preparing the fetal eye for vision by regulating retinal neuron number. They measured visceral cavity photon flux to be sufficient to activate certain regulating signals for retinal development in the fetal mouse eye [61]. Many studies cover the overall effect of UV radiation in pregnancy, but none indicate the effect of direct UV on the unborn child, let alone the fetal lens. Though UVA (320–400 nm) can penetrate to the dermis [62], it ultimately remains unknown how much UV effectively reaches the human fetal lens. Hence, the UV protective character of conserved Tryptophans in crystallins resembles an observation on the protein basis of these crystallin mutations only.

Mutations in two conserved Tryptophans were also found to be responsible for CC in β-crystallins like *CRYBB2*, in which mutations at p.Trp59Arg and p.Trp151Arg/Cys were reported to cause a significant change in the structural integrity and stability of β-crystallin, even more so than γ-crystallins [17,63,64,65,66] (Table 3). Xu et al. (2021) [64] identified a family with progressive cortical CC due to a Trp151Arg mutation in *CRYBB2*, displaying that the mutant protein increasingly misfolds, exposing hydrophobic side chains in the fourth Greek key, making it prone to aggregate. Interestingly, a complete prevention or reverse effect was described in vitro after lanosterol application to the pTrp151Arg mutant, posing a potential therapy option for CC patients with p.Trp151Arg mutations in *CRYBB2* [64]. However, children born with a dense CC may not be the target patient cohort for this approach.

To the best of our knowledge, we describe the first human nuclear CC caused by a novel de novo missense mutation at a highly conserved Tryptophan position, p.Trp131Arg, in the *CRYGC* gene, hypothesizing a similar disruption in the tertiary structure and solubility and stability dynamics in *CRYGC*. Functional assays would be necessary to provide conclusive evidence for pathogenicity of this specific variant.

## 5. Conclusions

We identified a novel de novo missense variant, c.391T>C, within exon 3 in *CRYGC* causing congenital nuclear cataract in a patient. Our findings expand the current understanding of the range of variants present in *CRYGC* and contribute crucial insight into the heterogeneity of inherited cataracts in the pediatric population.

## Figures and Tables

**Figure 1 ijms-24-16594-f001:**
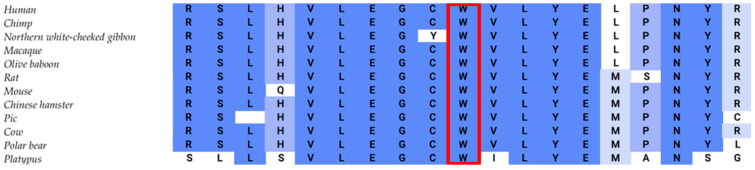
Amino acid conservation across species (https://www.ensembl.org/index.html (accessed on 4 September 2023)). The Tryptophan (W; marked red) affected by the identified variant is highly conserved among species. Dark blue indicates high, medium blue indicates moderate, light blue indicates minor and white indicates low conservation across species.

**Figure 2 ijms-24-16594-f002:**
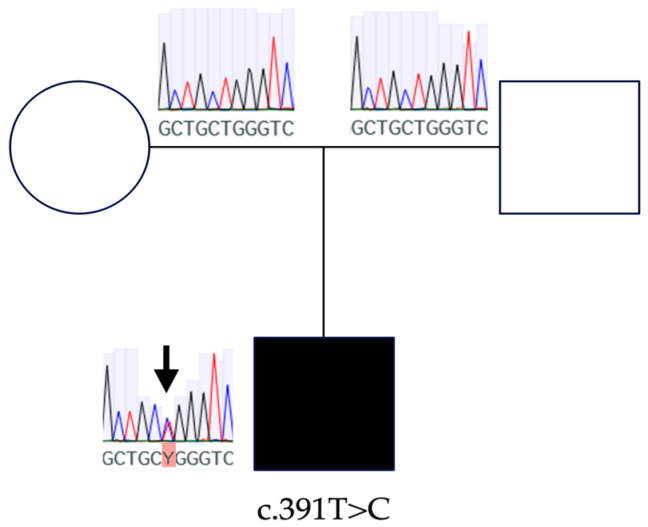
Sanger Sequencing variant verification.

**Table 1 ijms-24-16594-t001:** Disease-causing variant identified by WES.

Gene	*CRYGC*
cDNA	NM_020989.4:c.391T>C
Predicted amino acid change	p.Trp131Arg
Zygosity	het
gnomAD	n/a
Mode of inheritance	ad
Region	Exon 3
ACMG	pathogenic (PP3 + PM1 + PM2 + PS2) [23]

Acronyms: ACMG, American College of Medical Genetics and Genomics; het, heterozygous; n/a, not available; ad, autosomal dominant.

**Table 2 ijms-24-16594-t002:** Previously described disease-causing *CRYGC* variants.

Exon/ Intron	cDNA	Amino Acid Change	Coding Effect	Protein Domain	Phenotype	Reference
Exon 2	NM_020989.4:c.13A>C	p.Thr5Pro	missense	1st Greek key	Coppock-like CC	Heon et al. (1999) [25]; Berry et al. (2020) [26]
Exon 2	NM_020989.4:c.17T>C	p.Phe6Ser	missense	1st Greek key	Lamellar CC	Astiazaran et al. (2018) [27]
Exon 2	NM_020989.4:c.83C>T	p.Pro28Leu	missense	1st Greek key	Nuclear CC + microphthalmos + nystagmus	Jiao, et al. (2022) [28]
Exon 2	NM_020989.4:c.110G>C	p.Arg37Pro	missense	1st Greek key	CC NFS	Zhang et al. (2019) [29]
Exon 2	NM_020989.4:c.134T>C	p.Leu45Pro	missense	2nd Greek key	Non-syndromic CC	Gillespie et al. (2014) [10]; Fu et al. (2021) [30]
Exon 2	NM_020989.4:c.136T>G	p.Tyr46Asp	missense	2nd Greek key	Nuclear CC	Zhong et al. (2017) [31]; Fu et al. (2021) [30]
Exon 2	NM_020989.4:c.143G>A	p.Arg48His	missense	2nd Greek key	Nuclear pulverulent CC; unilateral CC + optic disc coloboma	Kumar et al. (2011) [32]; Sun et al. (2017) [33]
Exon 2	NM_020989.4: c.164A>G	p.Gln55Arg	missense	2nd Greek key	CC NFS	Karahan et al. (2021) [34]
Exon 2	NM_020989.4:c.173T>C	p.Leu58Pro	missense	2nd Greek key	CC NFS	Moon et al. (2021) [35]
Exon 2	NM_020989.4:c.233C>T	p.Ser78Phe	missense	2nd Greek key	CC + microcornea	Li et al. (2018) [36]
Exon 3	NM_020989.4:c.280G>A	p.Glu94Lys	missense	3rd Greek key	Unilateral total CC	Li et al. (2016) [37]
Exon 3	NM_020989.4:c.385G>T	p.Gly129Cys	missense	4th Greek key	CC NFS	Li et al. (2012) [38]; Xi et al. (2015) [39]
Exon 3	NM_020989.4:c.497C>T	p.Ser166Phe	missense	4th Greek key	Nuclear CC + microphthalmos	Prokudin et al. (2014) [40]; Zhong et al. (2017) [31]; Fan et al. (2020) [41]; Ma et al. (2016) [42]
Exon 3	NM_020989.4:c.502C>T	p.Arg168Trp	missense	4th Greek key	Lamellar/nuclear CC + peripupillary iris atrophy, nystagmus,	Santhiya et al. (2022) [43]; Gonzaez-Huerta et al. (2007) [44]; Devi et al. (2008) [45]
Exon 3	NM_020989.4:c.327C>A	p.Cys109Ter	nonsense	3rd Greek key	Nuclear CC	Yao et al. (2008) [46]
Exon 3	NM_020989.4:c.337C>T	p.Gln113Ter	nonsense	3rd Greek key	Nuclear CC	Li et al. (2016) [37]
Exon 3	NM_020989.4:c.382G>T	p.Glu128Ter	nonsense	3rd Greek key	Nuclear CC	Kandaswamy et al. (2020) [47]
Exon 3	NM_020989.4:c.402C>G	p.Tyr134Ter	nonsense	4th Greek key	CC NFS	Gillespie et al. (2014) [10]
Exon 3	NM_020989.4:c.403G>T	p.Glu135Ter	nonsense	4th Greek key	CC + microcornea	Patel et al. (2017) [48]
Exon 3	NM_020989.4:c.417C>G	p.Tyr139Ter	nonsense	4th Greek key	Total CC + microphthalmos	Reis et al. (2013) [49]
Exon 3	NM_020989.4:c.417C>A	p.Tyr139Ter	nonsense	4th Greek key	Nuclear CC + microcornea	Zhong et al. (2017) [31]
Exon 3	NM_020989.4:c.432C>G	p.Tyr144Ter	nonsense	4th Greek key	Nuclear CC	Zhong et al. (2017) [31]; Sun et al. (2017) [33]; Taylan Sekeroglu et al. (2020) [50]
Exon 3	NM_020989.4:c.470G>A	p.Trp157Ter	nonsense	4th Greek key	Nuclear CC + microcornea	Zhang et al. (2009) [51]; Kessel et al. (2021) [52]
Exon 3	NM_020989.4:c.471G>A	p.Trp157Ter	nonsense	4th Greek key	Nuclear CC + microcornea	Guo et al. (2012) [53]
Exon 3	NM_020989.4:c.505A>T	p.Arg169Ter	nonsense	4th Greek key	Nuclear CC	Zhong et al. (2017) [31]
Intron 1	NM_020989.4:c.10-1G>A		splicing		CC NFS	Zhuang et al. (2019) [54]
Exon 2	NM_020989.4:c.119_123dupGCGGC	p.Cys42AlafsTer63	frameshift	2nd Greek key	Zonular pulverulent CC	Ren et al. (2000) [55]
Exon 2	NM_020989.4:c.124delT	p.Cys42AlafsTer61	frameshift	2nd Greek key	Total CC ± microphthalmos	Kondo et al. (2013) [56]
Exon 2	NM_020989.4:c.130delA	p.Met44CysfsTer59	frameshift	2nd Greek key	Total CC + microcornea	Sun et al. (2017) [33]
Exon 2	NM_020989.4:c.157_161dup-GCGGC	p.Gln55ValfsTer50	frameshift	2nd Greek key	CC NFS	Reis et al. (2013) [49]
Exon 2	NM_020989.4:c.179delG	p.Arg60GlnfsTer43	frameshift	2nd Greek key	Nuclear CC	Berry et al. (2020) [26]
Exon 2	NM_020989.4:c.192delC	p.Asp65ThrfsTer38	frameshift	2nd Greek key	CC NFS	Fan et al. (2020) [41]
Exon 2	NM_020989.4:c.193delG	p.Asp65ThrfsTer38	frameshift	2nd Greek key	Nuclear CC	Zhong et al. (2017) [31]
Exon 3	NM_020989.4:c.320_321del-AA	p.Glu107GlyfsTer56	frameshift	3rd Greek key	Total CC	Rechsteiner et al. (2021) [9]
Exon 3	NM_020989.4:c.328_329del-CCinsT	p.Pro110SerfsTer37	frameshift	3rd Greek key	Lamellar CC	Ma et al. (2016) [42]
Exon 3	NM_020989.4:c.386_389dup-GCTG	p.Cys130TrpfsTer35	frameshift	4th Greek key	Nuclear CC ± microphthalmos	Zhou et al. (2022) [57]
Exon 3	NM_020989.4:c.394delG	p.Val132SerfsTer15	frameshift	4th Greek key	Total CC + microphthalmos	Peng et al. (2022) [13]
Exon 3	NM_020989.4:c.423delG	p.Arg142GlyfsTer5	frameshift	4th Greek key	Nuclear CC	Zhong et al. (2017) [31]
Exon 3	NM_020989.4:c.423dupG	p.Arg142AlafsTer22	frameshift	4th Greek key	Nuclear CC	Zhong et al. (2017) [31]
Exon 3	NM_020989.4:c.425_432dup	p.Leu145GlyfsTer5	frameshift	4th Greek key	Nuclear CC + microphthalmos + iris malformations	Fernández-Alcalde et al. (2021) [58]
Exon 3	NM_020989.4:c.438delG	p.Arg147GlyfsTer32	frameshift	4th Greek key	Nuclear CC	Fernandez-Alcade et al. (2021) [58]

Acronyms: CC, congenital cataract; NFS, not further specified.

**Table 3 ijms-24-16594-t003:** Previously described disease-causing point mutations in conserved Tryptophans of crystallins.

Gene	Exon	cDNA	Amino Acid Change	Coding Effect	Protein Domain	Phenotype	Reference
*CRYGD*	Exon 2	NM_006891.4:c.127T>C	p.Trp43Arg	missense	2nd Greek Key	Nuclear CC	Wang et al. (2011) [59]; Ji et al. (2013) [60]
*CRYBB2*	Exon 4	NM_000496.3:c.177G>C	p.Trp59Arg	missense	2nd Greek Key	Total CC	Santhiya et al. (2010) [63]; Zhao et al. (2017) [17]
*CRYBB2*	Exon 6	NM_000496.3:c.451T>C	p.Trp151Arg	missense	4th Greek key	Progressive CC	Xu et al. (2021) [64]
*CRYBB2*	Exon 6	NM_000496.3:c.453G>C	p.Trp151Cys	missense	4th Greek key	Progressive membranous CC	Chen et al. (2013) [65]; Zhao et al. (2017) [17]
*CRYBB2*	Exon 6	NM_000496.3:c.453G>T	p.Trp151Cys	missense	4th Greek key	Nuclear CC	Santhiya et al. (2004) [66]

Acronyms: CC, congenital cataract.

## Data Availability

The data presented in this study are available upon request from the corresponding author.

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
