# Peer review of "Novel *CRYGC* Mutation in Conserved Ultraviolet-Protective Tryptophan (p.Trp131Arg) Is Linked to Autosomal Dominant Congenital Cataract"

_ijms, 2023, doi:10.3390/ijms242316594_

Round 1
Reviewer 1 Report
Comments and Suggestions for Authors
In this manuscript the authors describe a novel CRYGC p.(Trp131Arg) mutation causing congenital cataract. The work is well conceived, the analysis is generally cogent, and the Discussion is mostly reasonable. The work would be greatly strengthened by including functional analysis of the mutation at the protein and perhaps cellular and tissue level, although this might not be necessary in a case report. Specific comments follow:
1. Introduction, p. 2, lines 57-58, "To date, the Online Mendelian Inheritance in Man 57 (OMIM) documented the identification of 49 loci and 37 genes associated with isolated 58 CC [13].": Is reference 13 the best reference for this statement?
2. Introduction, p. 2, lines 67-68, "α-crystallins are small heat shock proteins 67 (HSP20).": It would probably be best to remove '(HSP20)' or explain why it is there.
3. Materials..., p. 2, line 89, "The index patient was identified through the cataract genetic study.": Perhaps a bit more description of the study would help here.
4. Table 2. I think the list of CRYGC mutations here is incomplete. The authors might wish to check CAT-MAP for additional mutations.
5. Discussion, p. 6, lines 176-179, "Chen et al. (2009) 176 [18] revealed significant findings on the ability to effectively quench excited states through 177 electrostatic interactions of four highly conserved Tryptophans (Trp 43, Trp69, Trp131, 178 and Trp 157) on a protein basis, to be an evolved property of all γ -crystallins to maintain 179 the tertiary structure as a form of UV protection.": The authors might mention that since these, and most of the mutations included in table 2, are congenital or childhood cataracts UV quenching by tryptophans would probably have a very limited role in their pathophysiology.
6. Discussion, p. 6, lines 195-196, "Very few studies cover the effect of UV radiation on the unborn child during preg-195 nancy, all of which carry bias due to their observational character [58,59].": These reviews cover the overall effect of UV light on a pregnancy, not the direct effect of UV light on a fetus or fetal lens. Since UV light could not penetrate the skin, this should be minimal or absent, at least in terms of direct UV damage.
Reviewer 2 Report
Comments and Suggestions for Authors
The submission reports a novel mutation in a lens crystallin, CRYGC. It is one of the major gamma-crystallin expressed in the human lens. The mutation, W131R. This residue is part of the tryptophan corner (Aravind et al 2008) and because the two domains of CRYGC rely upon their interface for stability, this mutation would be expected to have a dramatic effect upon protein stability leading to its aggregation and the appearance of a dense nuclear cataract. Domain 1 is not as stable as Domain 2 in the gamma-crystallins, and mutations that affect domain 2 such as the W131R reported here will also affect domain 1 stability. There is little doubt in my mind that the W131R mutation is the cause of the cataract as strongly indicated by the pyloP score and ACMG rating.
The mutation is proposed to be spontaneous and on lines 166/167, it is claimed that the mutation was absent from both parents. This statement needs to be tested more rigorously as paternal germline mosaicism (de Manuel et al 2023) should be excluded before this can be left unchanged in the manuscript (See Frisk et al 2022). The authors have blood samples from both parents that will allow this additional important information to be provided.
With respect to the possibility of UV light penetrating the womb to influence the foetus, the Rao et al Nature 2013; Yang et al 2013 papers could deserve mentioning and perhaps the authors might consider the potential impact of the Greenberg et al 2023 paper on UV light exacerbating the aggregation potential of CRYGC R131R mutation whilst in the womb. Once born that might change, but before birth I think that this argument needs to be reassessed and relevant literature cited.
Frisk S, Wachtmeister A, Laurell T, Lindstrand A, Jäntti N, Malmgren H, Lagerstedt-Robinson K, Tesi B, Taylan F, Nordgren A. Detection of germline mosaicism in fathers of children with intellectual disability syndromes caused by de novo variants. Mol Genet Genomic Med. 2022 Apr;10(4):e1880. doi: 10.1002/mgg3.1880. Epub 2022 Feb 4. PMID: 35118825; PMCID: PMC9000944.
Marc de Manuel, Felix L Wu, Molly Przeworski (2022) A paternal bias in germline mutation is widespread in amniotes and can arise independently of cell division numbers eLife 11:e80008
Greenberg J, Gruner K, Rodney L, Struve J, Kang D, Cao Y, Lang R. Biologically Aware Lighting for Newborn Intensive Care. Res Sq [Preprint]. 2023 Jul 10:rs.3.rs-3120637. doi: 10.21203/rs.3.rs-3120637/v1. PMID: 37502905; PMCID: PMC10371081.
Reviewer 3 Report
Comments and Suggestions for Authors
In the manuscript the authors present a novel CRYGC mutation in a patient with cataract, these are some of my comments:
Please change “index” by proband
What about Genomic Evolutionary Rate Profiling?
Correct please legend of table 2
More in silico analysis is necessary.
A table or schema of comparative analysis of the amino acid through several species would be useful.
Comments on the Quality of English LanguageNo comments
Round 2
Reviewer 3 Report
Comments and Suggestions for Authors
No comments
Comments on the Quality of English LanguageNo comments
